# Best practices and practical strategies for co-designing virtual reality with Indigenous peoples: A scoping review protocol

Lillian Hung[1,2], Lenny Fisher[3], Jeffrey Wong◉[1,4*], Yong Zhao[1], Katrina Yuen[1], Lily Haopu Ren◉[1]

1 Innovation in Dementia and Aging Lab, The University of British Columbia, Vancouver, Canada, 2 School of Nursing, The University of British Columbia, Vancouver, Canada, 3 Ch'íyáqtel (Tzeachten First Nation), Chilliwack, Canada, 4 Department of Sociology and Anthropology, Trinity Western University, Langley, Canada

* jeffrey.wong@ubc.ca

## Abstract

Virtual reality (VR) is gaining traction in healthcare, education, and cultural sectors, from simulations in medical education to immersive museum experiences. Recently, VR has emerged as a powerful tool for Indigenous cultural preservation, language revitalization, and storytelling, offering immersive ways to safeguard knowledge and strengthen community connections. However, despite VR's potential to support Indigenous self-determination, little is known about the extent of Indigenous leadership, engagements, and settler-Indigenous collaborations in VR development. There is a critical need to examine how VR can be ethically and meaningfully co-designed with Indigenous communities to ensure cultural integrity, respect for Indigenous knowledge systems, and equitable participation in technological innovation. Thus, this scoping review aims to identify practical strategies and best practices for co-designing VR with Indigenous communities. In accordance with the JBI methodology, we will conduct a comprehensive search across seven electronic databases, including MEDLINE (EBSCOhost), Scopus, Web of Science, ACM Digital Library, IEEE Xplore, Compendex (Engineering Village), and ProQuest Dissertations and Theses Global (ProQuest). Google Scholar will also be searched for grey literature sources. Eligible studies will focus on Indigenous populations (Population) and fully immersive VR co-design (Concept) across various contexts. Studies that do not discuss the design process will be excluded. Two independent reviewers will conduct literature screening, data extraction, and analysis, with findings synthesized narratively and presented in a structured charting table. The results will be disseminated through a peer-reviewed journal publication and shared with relevant community partners to support knowledge translation and application.

**Data availability statement:** All relevant data from this study will be made available upon study completion.

**Funding:** This scoping review is supported by funding from AGE-WELL (Grant number: GR033585; https://agewell-nce.ca) awarded to LH. The sponsor does not play a role in the study design, data collection and analysis, decision to publish, or preparation of the manuscript.

**Competing interests:** The authors have declared that no competing interests exist.

## Introduction

Virtual reality (VR) technology is increasingly embraced by sectors beyond personal entertainment, including healthcare, education, cultural and language preservation. Defined as a three-dimensional (3D) computer-generated simulated environment [1], VR's immersive abilities make it ideal for both closely resembling real-life scenarios and creating fictional environments [2]. A systematic review on immersive VR in healthcare [3] found that current clinical applications primarily revolve around treatment or management of neurological and neurodevelopmental conditions, pain relief, exposure therapy for phobias, and other psychological conditions (e.g., depression [4] and unhealthy eating habits). For example, researchers investigated the effectiveness of VR cognitive training programs in post-stroke neuropsychological rehabilitation and found that participants in the VR intervention group showed significant improvements in attention and memory functions, but not the control group [5]. Fully immersive VR training programs also enhance dementia care by fostering empathy, improving caregivers' ability to recognize and respond to behavioural triggers, and supporting skill development through interactive, safe, and engaging learning environments [6]. VR has also been explored to promote social engagement and wellbeing for people living with dementia, both in acute and long-term care settings [7,8], and in the community [9]. Furthermore, multisensory VR, particularly when integrated with aromatherapy, holds potential for improving older adults' physical, psychological, cognitive well-being, and social engagement [10]. Beyond healthcare, VR has been implemented into higher education a scenario-based learning tool, particularly in facilitating simulations in some medical and engineering courses [11]. Meanwhile, in cultural heritage preservation, VR offers unique opportunities to visualize historical sites that no longer exist [12,13], digitize traditional art forms in 3D, such as Chinese paper cutting art [14], and create immersive museum or exhibition experiences that allow users to interact and engage with artifacts [15].

In Indigenous contexts, VR is increasingly recognized as a powerful tool for the preservation of Indigenous knowledge and culture [16]. It has been used to recreate sacred landscapes, revitalize Indigenous languages, and support cultural education [17]. However, the development of VR applications for Indigenous communities requires careful consideration to ensure that technology aligns with community priorities, respects Indigenous knowledge systems, and upholds cultural integrity.

Co-design is an emerging concept and is currently defined and used in various ways across health research [18]. It involves the meaningful engagement of research users throughout the planning and study phases of a research project [19]. It supports an authentic and self-determined process, where users are empowered to actively participate in the decision-making processes that impact themselves and their communities [20]. In the development of VR applications, a participatory approach is crucial to ensure the technology best aligns with user needs, values, traditions, cultural contexts, and ways of knowing [20]. One notable application for co-design is in Indigenous communities, where it can be used to drive changes in health interventions and policies, language preservation, and storytelling [18]. For co-design to be effective, Indigenous involvement should be purposeful and

respectful, guided by shared decision-making policies and cultural protocols. However, user involvement terminology—such as co-creation, co-design, and co-production—is often used interchangeably, leading to conceptual ambiguity [21–23]. Vargas et al. [24] distinguishes co-design as a participatory, multi-stakeholder approach spanning all phases of design, from problem identification to evaluation, while co-production primarily occurs during implementation to optimize available resources. In the context of co-designing VR experiences with Indigenous communities to preserve their culture, language, and storytelling, co-design involves diverse stakeholders—including Indigenous knowledge keepers and community members (primary users), educators, language and cultural preservation organizations, and Indigenous artists (secondary users), and policymakers, healthcare providers, researchers, and technology developers (tertiary users)—as equal partners throughout the design process [21]. While this inclusive approach fosters meaningful engagement and ensures that VR storytelling authentically reflects Indigenous traditions, this scoping review focuses on the most central group, Indigenous peoples, to explore best practices based on their lived experiences [25].

While a previous scoping review explored inclusivity in VR development, it has primarily identified barriers to participation rather than offering concrete best practices [26]. Under-representation of Indigenous perspectives was mentioned but specific strategies for effective collaboration remain largely absent from the literature. To bridge this gap, we will conduct a scoping review in accordance with the JBI methodology to explore practical strategies and best practices in co-designing VR with Indigenous peoples. Findings from this review will provide guidance for researchers, developers, and policymakers seeking to implement ethical and culturally responsive VR initiatives with Indigenous peoples.

## Review question

What practical strategies and best practices are effective in co-designing VR with Indigenous peoples?

## Inclusion criteria

### Participants

We will focus on studies engaging participants and/or partners who are self-reported as Indigenous. We will include studies co-designing VR with Indigenous partners from various demographic and background, as we acknowledge the heterogeneity and diversity among Indigenous people from different communities. Papers focusing on co-designing VR without any Indigenous partners and/or participants will be excluded.

### Concept

This review will include studies that involves co-designing, co-producing, or co-creating fully immersive VR videos and programs with Indigenous partners and/or participants. We will include studies did not explicitly state co-design as the research method yet active involve and engage with Indigenous communities in the VR design and production process. We will also include studies of Indigenous-led VR development. Mixed reality (MR) merges the physical and digital environments, enabling real-time interaction between virtual and real-world elements, typically through head-mounted displays [27]. Due to its immersive and interactive qualities, this scoping review includes studies that integrate MR alongside VR. Studies involving co-design of non-VR programs, such as augmented reality (AR), will be excluded.

### Context

This review will include co-designing VR in various contexts regardless of countries and cultural or ethnic backgrounds. We will include studies that co-design VR in individual places, community, locations for educational, cultural and religious activities and various healthcare settings (e.g., hospitals, long-term care homes). We will also include studies for different purposes, such as education, communication, cultural preservation and care delivery. We will include co-designed VR for care delivery to population with different cultural background and a variety of conditions, and we will also include co-designed VR used in combination with other care programs (e.g., multi-sensory stimulation).

## Types of evidence sources

This scoping review will include a variety of study designs, encompassing both experimental and quasi-experimental approaches, such as randomized controlled trials, non-randomized controlled trials, before-and-after studies, and interrupted time-series analyses. It will also consider analytical observational studies, including prospective and retrospective cohort studies, case-control studies, and analytical cross-sectional studies. Additionally, descriptive observational designs, such as case series, individual case reports, and descriptive cross-sectional studies, will be included.

Qualitative studies that focus on qualitative data will be incorporated, utilizing methodologies such as phenomenology, grounded theory, ethnography, qualitative description, action research, and feminist research. Systematic reviews that align with the inclusion criteria may be included, depending on their relevance to the research question. Text and opinion papers will also be considered for inclusion in this scoping review.

## Methods

The proposed scoping review will be conducted in accordance with the JBI methodology for scoping reviews [28] and will be reported in line with the Preferred Reporting Items for Systematic reviews and Meta-Analyses extension for Scoping Reviews (PRISMA-ScR) [29]. Conducting a scoping review is well-suited to our research question, as it allows for a broad exploration of the Indigenous VR co-designing field and evaluates the scope of research activity to guide both practice and future studies. This protocol was developed with guidance from the JBI Scoping Review Methodology Group [30]. Study screening, data extraction, and analysis will be conducted between March and June 2025.

## Search strategy

The search aims to identify both published and unpublished studies. An initial search in MEDLINE and Google Scholar (first 50 results) was conducted to locate relevant articles (S1 Appendix). Keywords and index terms extracted from article titles and abstracts were used to develop a comprehensive search strategy for English-language databases, including MEDLINE (EBSCOhost), Scopus, Web of Science, ACM Digital Library, IEEE Xplore, Compendex (Engineering Village), and ProQuest Dissertations and Theses Global (ProQuest). This strategy, incorporating all identified keywords and index terms, will be adapted for each database and information source. Additionally, reference lists of all included sources will be reviewed to identify additional relevant studies. Given the scoping review approach, we aim to capture all potential articles. For instance, due to variations in the stages of involvement of Indigenous peoples, we will collect studies related to co-design and co-creation activities and manually assess them during the screening process.

Studies in any language with English abstracts will be considered, with a cut-off date of February 2025. However, only full-text articles in English and Chinese will be screened in full, based on the language proficiency of the research team. Unpublished studies and grey literature will be sourced through Google Scholar. Conference abstracts will be used to establish contact with study authors, potentially enabling access to full-text papers.

## Evidence screening and selection

Following the search, all identified evidence sources will be collected and imported into the systematic review software, Covidence (Veritas Health Innovation, Melbourne, Australia), where duplicate records will be removed. Two independent reviewers will then screen titles and abstracts based on the inclusion criteria. Full texts of potentially relevant sources will be retrieved, with citation details imported into Covidence for further assessment. Two reviewers will independently evaluate the full texts against the inclusion criteria, documenting and reporting exclusion reasons for studies that do not qualify. Any disagreements between reviewers will be resolved through group discussion. The final scoping review will present the outcomes in detail, accompanied by a Preferred Reporting Items for Systematic Reviews and Meta-analyses extension for scoping review (PRISMA-ScR) flow diagram to illustrate the study selection process [29].

## Data extraction

Data will be extracted from selected articles by two independent reviewers. The Garrard Matrix method will be used to input data into a structured spreadsheet, capturing key details such as participants, study designs, practical strategies, practices, and findings relevant to the review questions [31] (S2 Appendix). In line with JBI methodology, a pilot test will be conducted using three full-text articles to ensure the reliability of the extraction tool, with results mapped for consistency. Since this scoping review focuses on mapping the literature rather than critically appraising evidence, study quality and methodological assessment will not be performed. The draft spreadsheet will be refined as needed based on team discussions, with any changes documented in the full review write-up. If required, study authors will be contacted to obtain missing or additional data.

## Data analysis and presentation

The extracted data will be presented in tabular format, supplemented with a narrative summary that aligns the findings with the study objectives and research questions. A thematic narrative mapping summary will be provided alongside the tabulated results to structure the information. This comprehensive review will present both qualitative and quantitative data.

## Inclusivity in global research

In accordance with PLOS' guidelines on best practices in global research, we have completed the Inclusivity in Global Research questionnaire, which addresses ethical, cultural, and scientific considerations when conducting research involving Indigenous peoples (S4 Appendix).

## Patient and public involvement

Our Indigenous partners, Lenny Fisher (Ch'íyáqtel) and Lynn Jackson (Métis), were involved in the conception and planning of the scoping review and will be engaged in the data analysis stage of the scoping review. They are Indigenous partners of the UBC Innovation in Dementia and Aging (IDEA) Lab, contributing their lived experiences to research and fostering an inclusive environment to ensure our research aligns with the needs of Indigenous communities. In this scoping review, they will actively participate in the study planning and analysis by contributing to group discussions. Their experiences regarding practical strategies and practices of VR filming and story-telling co-designing will be shared and communicated to the entire review group. Team discussions will be arranged based on their preferred schedules. They will also be invited to staff huddles and meetings to share key findings. To acknowledge their expertise and contributions, they will be invited to be co-authors of the scoping review report article.

## Ethics and dissemination

As this scoping review draws on published and unpublished articles, research ethics board approval is not required. The review will be submitted to a peer-reviewed, open-access journal to ensure broad accessibility. Additionally, a concise one-page summary will be created to facilitate knowledge translation and mobilization. Findings will be shared through staff huddles, community meetings, and relevant stakeholder discussions, including those involving Indigenous organizations and policymakers. Our goal is to provide evidence-based insights to support the co-design of VR programs with Indigenous communities, offering guidance for culturally informed and effective implementation across various settings.

## Supporting information

**S1 Appendix. Search strategy for MEDLINE (EBSCOhost).**
(DOCX)

**S2 Appendix. Data extraction instrument.**
(DOCX)

**S3 Appendix. PRISMA-P (Preferred Reporting Items for Systematic review and Meta-Analysis Protocols) 2015 checklist: recommended items to address in a systematic review protocol.**
(DOCX)

**S4 Appendix. PLOS questionnaire on inclusivity in global research.**
(DOCX)

## Acknowledgments

We would like to acknowledge the support from Katherine Miller, Reference Librarian at the University of British Columbia, who was consulted in an early version of our search strategy.

## Author contributions

**Conceptualization:** Lillian Hung, Jeffrey Wong.

**Funding acquisition:** Lillian Hung.

**Methodology:** Jeffrey Wong, Yong Zhao, Lily Haopu Ren.

**Project administration:** Jeffrey Wong.

**Supervision:** Lillian Hung.

**Writing – original draft:** Jeffrey Wong, Yong Zhao, Katrina Yuen, Lily Haopu Ren.

**Writing – review & editing:** Lillian Hung, Lenny Fisher, Jeffrey Wong, Yong Zhao, Katrina Yuen, Lily Haopu Ren.

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
