## [Decision Letter · Decision Letter 0]

11 Apr 2025

PONE-D-25-12586Practical strategies and best practices in co-designing virtual reality with Indigenous peoples and communities: A Scoping review protocolPLOS ONE

Dear Dr. Wong,

Thank you for submitting your manuscript to PLOS ONE. After careful consideration, we feel that it has merit but does not fully meet PLOS ONE’s publication criteria as it currently stands. Therefore, we invite you to submit a revised version of the manuscript that addresses the points raised during the review process.

We look forward to receiving your revised manuscript.

Kind regards,

Diogo Guedes Vidal, PhD

Academic Editor

PLOS ONE

Journal Requirements:

Reviewers' comments:

Reviewer's Responses to Questions

**Comments to the Author**

1. Does the manuscript provide a valid rationale for the proposed study, with clearly identified and justified research questions?

Reviewer #1: Yes

2. Is the protocol technically sound and planned in a manner that will lead to a meaningful outcome and allow testing the stated hypotheses?

Reviewer #1: Yes

3. Is the methodology feasible and described in sufficient detail to allow the work to be replicable?

Reviewer #1: No

4. Have the authors described where all data underlying the findings will be made available when the study is complete?

Reviewer #1: Yes

5. Is the manuscript presented in an intelligible fashion and written in standard English?

Reviewer #1: Yes

6. Review Comments to the Author

You may also provide optional suggestions and comments to authors that they might find helpful in planning their study.

Reviewer #1: In terms of method, there is a lack of the following items required by Plos One in the study planning: hypothesis, sample size calculation, type of data and statistical analyses to be used and where and when the data will be made available.

Line 142: What will be the analysis period? Please, include.

Line: 189: I suggest improving the wording by stating that these will be published articles. The word "available" gives the impression that only open access articles will be analyzed.

7. PLOS authors have the option to publish the peer review history of their article (what does this mean? ). If published, this will include your full peer review and any attached files.

**Do you want your identity to be public for this peer review?** For information about this choice, including consent withdrawal, please see our Privacy Policy .

Reviewer #1: **Yes: ** Rodrigo Rodrigues de Freitas

---

## [Author Response · Author response to Decision Letter 1]

3 May 2025

Academic Editor Comments - Response

1. We have revised our manuscript to align with the PLOS ONE formatting guidelines, including style and file naming conventions, as outlined in the templates provided. All files have been renamed and formatted accordingly.

2. We have completed the PLOS questionnaire on inclusivity in global research and uploaded it as Supporting Information (S4 Appendix) as per journal policy. We appreciate this important step in promoting transparency and ethical engagement in Indigenous research contexts.

3. We have included the following Data Availability Statement in the submission form: “All relevant data from this study will be made available upon study completion.”

4. Two authors (JW and KY) have thoroughly reviewed the reference list to ensure accuracy and completeness. We confirm that as of April 28, 2025, no retracted articles have been cited.

Reviewer 1 Comments - Response

1. Thank you for your comment. As this is a protocol for a scoping review, it does not involve primary data collection or hypothesis testing and therefore does not require a sample size calculation or statistical analysis plan. Scoping reviews are designed to “determine the scope or coverage of a body of literature on a given topic and give clear indication of the volume of literature and studies available as well as an overview (broad or detailed) of its focus” (Munn et al., 2018). Our data availability statement will be submitted through the submission system.

Reference:

Munn, Z., Peters, M. D. J., Stern, C., Tufanaru, C., McArthur, A., & Aromataris, E. (2018). Systematic review or scoping review? Guidance for authors when choosing between a systematic or scoping review approach. BMC Medical Research Methodology, 18(1), 143. https://doi.org/10.1186/s12874-018-0611-x

2.Thank you for the comment. As this is a scoping review, we interpret “analysis period” to refer to the expected timeframe during which study screening, data extraction, and synthesis will occur. We have added the following sentence to our manuscript to clarify this timeline:

Line 135: “Screening, data extraction, and analysis will be conducted between March and June 2025.”

3. We agree with this suggestion and have replaced “publicly available” with “published and unpublished” to better reflect the scope of our review. The revised sentence now reads:

Line 203: “As this scoping review draws on published and unpublished articles, research ethics board approval is not required. The review will be submitted to a peer-reviewed, open-access journal to ensure broad accessibility.”

---

## [Editor Report · Decision Letter 1]

7 May 2025

Best practices and practical strategies for co-designing virtual reality with Indigenous peoples: A scoping review protocol

PONE-D-25-12586R1

Dear Dr. Wong,

We’re pleased to inform you that your manuscript has been judged scientifically suitable for publication and will be formally accepted for publication once it meets all outstanding technical requirements.

Kind regards,

Diogo Guedes Vidal, PhD

Academic Editor

PLOS ONE
---

## [Editor Report · Acceptance letter]

PONE-D-25-12586R1

PLOS ONE

Dear Dr. Wong,

I'm pleased to inform you that your manuscript has been deemed suitable for publication in PLOS ONE. Congratulations! Your manuscript is now being handed over to our production team.

Kind regards,

on behalf of

Dr. Diogo Guedes Vidal

Academic Editor

PLOS ONE